# PCR-Based Strategy for Introducing CRISPR/Cas9 Machinery into Hematopoietic Cell Lines

**DOI:** 10.3390/cancers15174263

**Published:** 2023-08-25

**Authors:** Elisa González-Romero, Cristina Martínez-Valiente, Gema García-García, Antonio Rosal-Vela, José María Millán, Miguel Ángel Sanz, Guillermo Sanz, Alessandro Liquori, José Vicente Cervera, Rafael P. Vázquez-Manrique

**Affiliations:** 1Hematology Research Group, Instituto de Investigación Sanitaria La Fe, 46026 Valencia, Spain; elisa_gonzalez@iislafe.es (E.G.-R.); cristina_martinez@externos.iislafe.es (C.M.-V.); antonio.rosal@uca.es (A.R.-V.); miguel.sanz@uv.es (M.Á.S.); sanz_gui@gva.es (G.S.); alessandro_liquori@iislafe.es (A.L.); 2CIBERONC, 28029 Madrid, Spain; 3Laboratory of Molecular, Cellular and Genomic Biomedicine, Instituto de Investigación Sanitaria La Fe, 46026 Valencia, Spainmillan_jos@gva.es (J.M.M.); 4CIBERER, 46010 Valencia, Spain; 5Joint Unit for Rare Diseases IIS La Fe-CIPF, 46012 Valencia, Spain; 6Biomedicine, Biotechnology and Public Health Department, Cádiz University, 11002 Cádiz, Spain; 7Institute of Research and Innovation in Biomedical Sciences of Cadiz (INIBICA), 11009 Cádiz, Spain; 8Hematology Department, Hospital Universitari i Politècnic La Fe, 46026 Valencia, Spain; 9Genetics Unit, Hospital Universitari i Politècnic La Fe, 46026 Valencia, Spain

**Keywords:** leukemia, hard-to-transfect cell lines, sgRNA constructs, fusion PCR

## Abstract

**Simple Summary:**

We used PCR-generated small CRISPR constructs to edit two genes (IDH2 and MYBL2) in hard-to-transfect hemopoietic cells, which are central to the progression of the devastating disease known as acute myeloid leukemia LMA (AML). MYBL2 is a transcription factor; when AML patients show an altered expression of this factor, an adverse prognostic value is involved. IDH2 is particularly interesting because it encodes isocitrate dehydrogenase 2, an enzyme of the citric acid cycle, which, when mutated, produces a different phenotype in AML patients. Hence, our system provides a way to produce CRISPR constructs to easily target genes within mammalian cells, and it provides a model which can be used to study AML mechanisms in vitro.

**Abstract:**

Acute myeloid leukemia is a complex heterogeneous disease characterized by the clonal expansion of undifferentiated myeloid precursors. Due to the difficulty in the transfection of blood cells, several hematological models have recently been developed with CRISPR/Cas9, using viral vectors. In this study, we developed an alternative strategy in order to generate CRISPR constructs by fusion PCR, which any lab equipped with basic equipment can implement. Our PCR-generated constructs were easily introduced into hard-to-transfect leukemic cells, and their function was dually validated with the addition of MYBL2 and IDH2 genes into HEK293 cells. We then successfully modified the MYBL2 gene and introduced the R172 mutation into the IDH2 gene within NB4 and HL60 cells that constitutively expressed the Cas9 nuclease. The efficiency of mutation introduction with our methodology was similar to that of ribonucleoprotein strategies, and no off-target events were detected. Overall, our strategy represents a valid and intuitive alternative for introducing desired mutations into hard-to-transfect leukemic cells without viral transduction.

## 1. Introduction

Clustered regularly interspaced short palindromic repeats (CRISPR) have been identified in archaea and eubacteria; they consist of repetitive DNA sequences that, along with CRISPR-associated (Cas) proteins, serve as a natural adaptive defense against prokaryotic viral infection [1,2,3,4]. 

The main components of the CRISPR/Cas system have been adapted to modify the genomes of a variety of cells and organisms [5,6,7,8,9], leading to a revolution in the development of novel, personalized disease models. As reviewed by González-Romero et al. [10], CRISPR technology has facilitated the generation of several hematopoietic cell models. However, their poor transfection efficiency is still a considerable limitation. Although viral vectors currently represent the favored method for introducing CRISPR into cells [11,12,13,14], viral transduction has some drawbacks. For instance, adeno-associated viruses (AAVs) are constrained by their packaging capacity (of only 4.7 kb) [15], requiring the separate co-delivery of CRISPR/Cas9 transgenes through two different AVV vectors, which reduces efficiency [16]. Furthermore, the viral capsid of adenoviruses (AdV) may trigger acute immune responses within recipient models [17]. Finally, the strong integration capacity of lentiviruses and the consequent constitutive CRISPR/Cas9 expression may increase non-target modifications, which is a major barrier to the clinical implementation of this technology [16,17]. 

Acute myeloid leukemia (AML) is a heterogeneous disease characterized by the clonal expansion of myeloid precursors, resulting in impaired hematopoiesis and bone marrow abnormalities [18]. Among other genetic alterations, mutations in the isocitrate dehydrogenase 2 enzyme gene (IDH2) have been linked to distinctive gene expression patterns and epigenetic changes among AML patients; they have also been associated with specific clinical outcomes [19]. Specifically, two recurrent mutations have been reported for *IDH2*—IDH2^R140^ and IDH2^R172^, the latter being suggested as a diagnostic biomarker for the molecular subclassification of AML patients [20]. Studying the effects of these mutations through CRISPR-generated cell and animal models may provide insights into novel therapeutic approaches for AML. 

In this study, we develop a reliable strategy to efficiently introduce CRISPR/Cas9 technology into the hard-to-transfect NB4 and HL60 hematological cells. Firstly, we design an intuitive fusion PCR protocol to assemble small constructs that express single guide RNAs (sgRNAs), including the enhanced green fluorescent protein (EGFP) reporter for visualizing transfection and/or isolating single cells. Then, we generate NB4 and HL60 cell lines that constitutively express Cas9 (henceforth referred to as NB4-Cas9 and HL60-Cas9, respectively), which we transfect with our constructs to model the IDH2^R172^ mutation. Next, we evaluate the effectiveness and versatility of this approach by successfully targeting *MYBL2* in both cell lines. The *MYBL2* gene encodes a transcription factor that is pivotal in regulating the cell cycle, survival, and differentiation [21]; its dysregulation is reported in various cancers, including AML [22]. Finally, we compare the efficiency of our constructs with ribonucleoprotein complexes through DNA-targeted deep sequencing to evaluate the efficiency and to detect potential off-target events. Overall, this study introduces a novel and cost-effective gene editing strategy, exemplified by the targeting of *IDH2* and *MYBL2* genes in hard-to-transfect hematopoietic cell lines.

## 2. Materials and Methods

### 2.1. Materials Used in This Work

#### 2.1.1. ssODN Template to Induce Homologous Recombination

To introduce the R172K mutation, a single-stranded oligo DNA nucleotide (ssODN) was designed to include 35 nt homology arms, the *IDH2*^R172K^ point mutation, and six silent mutations to prevent hCas9 reiterated cleavage: 5′ CCA CGC CTA GTC CCT GGC TGG ACC AAG CCC ATC ACC ATT GGC AGG CAC GCC CAT GGC GAC CAG GTA GGC CAG GGT GGA GAG GGG AT 3′. The sg*IDH2*_2 PAM silent change introduces a new cleavage site for the restriction enzyme HhaI (New England BioLabs, Ipswich, MA, USA). For the sg*IDH2*_1 PAM, it was impossible to modify the essential sequence; so, we introduced seven silent changes into the sg*IDH2*_1 sequence. The ssODN was synthesized as an UltramerTM DNA oligonucleotide (IDTDNA, Coralville, IA, USA).

#### 2.1.2. sgRNAs Used to Induce Cuts in DNA

The nucleotide sequence adjacent to the target region in *IDH2* was sequenced from NB4 DNA cells using Sanger sequencing to ensure that there were no single nucleotide polymorphisms (SNPs) present to impede the homologous recombination events; the sequence was subsequently input into CHOPCHOP v3 (chopchop.cbu.uib.no) to predict possible sgRNAs [23] (Sanger sequencing was carried out by Genomic Platform, IIS La Fe, Valencia, Spain). Two guides, sg*IDH2*_1 and sg*IDH2*_2, were chosen (Appendix A). For *MYBL2*, we selected a sgRNA in intron 3, according to criteria previously established in our laboratory. 

#### 2.1.3. PX458-sgMYBL2 Vector

The PX458 plasmid (Addgene 48138) [24] with the sg*MYBL2* insertion was produced in our laboratory following the instructions given by the authors [24]. 

#### 2.1.4. Cell Lines Used in This Work

HEK293: this is an immortalized cell line commonly used in scientific research. It was derived from human embryonic kidney (HEK) cells and is widely used for a variety of applications, including protein expression, drug testing, and gene editing experiments.

HEK293T: these cells are derived from HEK293 but possess the SV40 large T antigen, allowing them to generate recombinant proteins using plasmid vectors that carry the SV40 promoter.

NB4: this is a human cell line derived from acute promyelocytic leukemia that is widely used in cancer research. It is difficult for it to be transfected due to its low transfection efficiency, which hinders the introduction of foreign genes into the cells for experimental purposes.

HL60: this is also a human cell line derived from acute promyelocytic leukemia; it is commonly employed in cancer research. As with NB4, it is very difficult for this cell line to be transfected with nucleic acids of a large size.

#### 2.1.5. Primers used in this work 

All primers used in this work, for PCR, homologous recombination, library preparation, etc., are listed and described in Appendix A.

### 2.2. Methods

#### 2.2.1. Nucleic Acids Handling and Analysis

##### Cloning the sgRNA Cassette to Create pEGR1 Vector

In detail, to clone the sgRNA cassette (pU6 promoter, sgRNA scaffold, and terminator) in pEGFP-N1, primers with AflII restriction sites were used to amplify the sgRNA cassette from the PX458 plasmid. The pEGFP-N1 and sgRNA cassettes were briefly digested with AflII (Thermo Fisher Scientific, Waltham, MA, USA). The digested vector was dephosphorylated with alkaline phosphatase (New England Biolab, Ipswich, MA, United States). Both products were then purified using the QUIAquick^®^ PCR Purification Kit (QIAGEN, Hilden, Germany), ligated by T4 DNA ligase (Thermo Fisher Scientific, Waltham, MA, USA) at RT and transformed using One Shot^®^ TOP10 Electrocomp™ *E. coli* (Invitrogen, Waltham, MA, USA). PCR was used to screen for positive transformants. Finally, the cloned pEGFP-pU6-sgRNA reporter expression cassettes (which we named pEGR1) were purified using the QIAprep^®^ Spin Miniprep Kit (Qiagen, Venlo, The Netherlands) and sequenced by Sanger sequencing (Sanger sequencing was carried out by Genomic Platform, IIS La Fe, Valencia, Spain).

##### Creating the sgRNA Constructs by Fusion PCR

Different primer combinations were used to create the constructs that encode the sgRNAs against *IDH2* or *MYBL2*. All PCRs were carried out using Phusion high-fidelity polymerase (Thermo Fisher Scientific, Waltham, MA, USA), using pEGR1 as a template. We phosphorylated the primers with T4 polynucleotide kinase (New England Biolab) to generate the phosphorylated sg*MYBL2* construct (sg*MYBL2*-P). Amplicons were purified with the MinElute^®^ PCR Purification Kit (Qiagen, Venlo, The Netherlands) prior to transfection. A schematic representation of the fusion PCR-based generation of pU6-sgRNA and EGFP-U6-sgRNA is detailed in the results section. 

##### PCR-Amplification of the Cas9 Cassette

The entire humanized Cas9 was amplified from the hCas9 vector (Addgene plasmid ID: 41815) [25] by PCR, using Phusion high-fidelity DNA polymerase (Thermo Fisher Scientific, Waltham, MA, USA) with specific primers, and the amplicons were purified using the MinElute^®^ PCR Purification Kit prior to the cellular transfections.

##### Analysis of DNA Editing Efficiencies

Total genomic DNA was extracted 48 h after transfection, and target-specific cleavage sites were amplified by PCR, using specific primers for each locus (i.e., the sites of the theoretical cut by the Cas9 nuclease in *IDH2* and *MYBL2*). The PCRs were purified on agarose with the E.Z.N.A.^®^ Gel Extraction Kit (Omega Bio-Tek, Biel/Bienne, Switzerland). The efficiency of the non-homologous end joining (NHEJ) DNA repair pathway was evaluated using the T7 endonuclease I (T7-EI) assay (New England Biolabs, Ipswich, MA, USA) [26], which detects Cas9-induced mutations by cutting the DNA at mismatched nucleotides. The T7-EI assay was conducted as described in the literature [26]. Briefly, the T7-EI assay, also called the surveyor nuclease assay, is a technique used to identify and detect mutations in DNA. It is commonly used in genomics research to study genetic variations, such as single nucleotide polymorphisms (SNPs) or small insertions/deletions (indels). The assay works by cleaving mismatched DNA strands, created through annealing wild-type and mutant DNA, with the T7-EI nuclease enzyme. We used the T7-EI assay to characterize the CRISPR-induced changes in DNA. When the CRISPR cuts a given locus in gDNA and is then repaired by NHEJ repair, the resulting sequence differs from the wild type. Then, we can amplify this region, denature it with heat, and reanneal it back. Because some PCR products will wear mismatches, these will be identified and cut by the T7-EI nuclease. The products of these reactions can then be analyzed by gel electrophoresis or other methods to quantify the efficiency of the CRISPR guides.

Alternatively, the efficiency of the homologous direct repair (HDR) of DNA is evaluated with restriction fragment length polymorphism (RFLP) analysis. In this case, the DNA mutations introduced by the ssODN included the cleavage site of the HhaI restriction enzyme (New England BioLabs, Ipswich, MA, USA). Incubation of the purified amplicons with the HhaI enzyme was carried out following the manufacturer’s protocol. The T7E-I and RFLP products were determined using separation by 10% polyacrylamide electrophoresis, followed by staining with SYBR™ Safe DNA Gel Stain (Thermo Fisher Scientific, Waltham, MA, USA). ImageJ software (https://imagej.nih.gov/ij/index.html) was used to estimate the efficiency of NHEJ or HDR repair by measuring the integrated intensity of the undigested and digested end products.

##### Inference of CRISPR Edits (ICE) Analysis

The ICE analysis is a computational method used to determine the specific genomic changes induced by CRISPR-Cas9 gene editing. It takes sequencing data from targeted genomic regions and compares them to the reference sequence in order to identify any alterations introduced by the CRISPR-Cas9 system. The ICE analysis allows the assessment of the efficiency and accuracy of CRISPR-based gene editing experiments and provides valuable insights into the editing outcomes at the molecular level. Hence, to conduct this analysis, we amplified, by PCR, the regions targeted by CRISPR in the *IDH2* and *MYBL2* genes, using the Taq Polymerase PCR Master Mix (Thermo Fisher Scientific, Waltham, MA, USA). The amplicons were analyzed by Sanger sequencing (carried out by STAB VIDA, Caparica, Portugal), and the resulting .ab1 files were examined through the ICE analysis tool (https://ice.synthego.com/#/ (accessed on 12 March 2022)). Unfortunately, the ICE algorithm did not permit the analysis of HDR in the samples edited with two sgRNAs.

##### On- and Off-Target Analysis Using Next-Generation Sequencing

Cas-OFFinder (http://www.rgenome.net/cas-offinder/ (accessed on 23 March 2021)) [27] was used to identify the potential off-targets that differed from sg*IDH2*_1 and sg*IDH2*_2 by up to three mismatches (Table 1). The samples used for the RFLP analyses were dually used as templates in a two-step PCR strategy to analyze the on-targets and 16 potential off-targets. In the first step, the PCR primers for each locus contained an adapter sequence. The amplicons were purified using AMPure Beads (BD Biosciences, Franklin Lakes, NJ, USA), and in the second step, they were re-amplified with primers containing an adapter sequence that overlapped the first primers and an index sequence in the reverse primers. The final PCR products underwent a second round of purification using the AMPure Beads prior to library construction. Following the manufacturer’s protocol, the purified PCR products were pooled in equimolar amounts and sequenced on an Illumina MiSeq instrument with a MiSeq Reagent Kit v2 Micro (500 cycles). The deep sequencing data were analyzed through CRISPResso2 (https://crispresso.pinellolab.partners.org/submission (accessed on 20 April 2022)) [28] to evaluate the editing efficiency and possible off-target effects using default parameters. 

#### 2.2.2. Cell Culture 

##### Culture of Commercial Cell Lines

The HEK293T and HEK293 cells were cultured in DMEM (1X; Thermo Fisher Scientific), supplemented with 0.5% penicillin/streptomycin (100X Solution; Biowest, Minneapolis, MN, USA) and 10% fetal bovine serum (FBS; Thermo Fisher Scientific). The cultures were maintained in a Thermo 3310 Steri-Cult CO2 Double Incubator Unit4 (Thermo Fisher Scientific Waltham, MA, USA), at 37 °C with 5% CO_2_. The NB4 and HL60 cells were cultured in RPMI Medium 1640 (Gibco, Thermo Fisher Scientific Waltham, MA, USA), supplemented with 1% penicillin/streptomycin (100× Solution) and 10% FBS. The NB4-Cas9 and HL60-Cas9 medium was additionally supplemented with 0.2 µg/mL of puromycin. The cells were maintained in a humidified incubator at 37 °C with 5% CO_2_. 

##### Assembling Ribonucleoprotein Complexes

To form the ribonucleoprotein (RNP) ensembled complexes, the corresponding Alt-R^®^ CRISPR/Cas9 crRNA and tracrRNA (IDT) were hybridized to form the sgRNA according to the manufacturer’s protocol. Prior to nucleofection, 17 µg of purified recombinant S. pyogenes Cas9 nuclease (IDT) was added to 20.3 µg of each sgRNA, followed by incubation at room temperature for 10 min, to form the RNP complexes. 

##### Nucleofection of Cell Lines

For the nucleofection experiments, we used Cell Line Nucleofector Solution Kit V (Lonza Basel, Switzerland); 2 × 10^5^ cells and pMAX plasmid were transfected as a positive control. We followed the Lonza protocol for Kit V.

To transfect sgRNA, constructs in NB4-Cas9 and HL60-Cas9 were necessary to optimize the construct concentration. We tried 800 ng and 1500 ng of the sg*IDH2* and sg*MYBL2* constructs.

For the *IDH2*^R172^ introduction, we used 100 µM since the 10 µM optimized concentration in the HEK293 cells was not effective.

##### Transfection of CRISPR Constructs

For the CRISPR transfections in HEK293, we used Lipofectamine™ 3000, following the instructions of the manufacturer (Invitrogen); 0.9 × 10^5^ cells and pMAX plasmid were used as a transfection control. We tested different combinations of the hCas9 and sg*MYBL2* constructs to optimize target-specific cleavage (see Results). For *IDH2* editing, 35 ng of both guides (17.5 ng of each guide) were co-transfected with 250 ng of hCas9 vector. For the R172 mutation editing experiments, 10 µM of ssODN was used.

##### Generation of the NB4 and HL60 Cells Constitutively Expressing Cas9 

The day before transfection, the HEK293T cells were plated at a density of 3 × 10^6^ in a 10 cm culture dish. The cells were transfected with lentiCRISPR v2 (Addgene plasmid ID:52961) [29], which contains the Cas9 nuclease and puromycin resistance gene from Streptococcus pyogenes, and two packaging plasmids (pPAX2 and pMD2.G), using the calcium chloride method. To perform this, we mixed the DNA with calcium chloride and added it to the cultured cells. The supernatant containing the lentivirus was collected 48 h post-transfection, filtered through a 0.45 µm nitrocellulose filter, and stored at −80 °C until further use. 

For lentiviral transduction, 5 × 10^5^ NB4 and HL60 cells were infected with 100 µL of the lentivirus supernatant containing polybrene at a final concentration of 4 µg/mL. After 24 h, the transduced cells were centrifuged and resuspended and cultured in a medium containing 0.6 µg/mL puromycin (InvivoGen, San Diego, CA, USA) for several days. To isolate the cells with viral integration, DNA was extracted to detect the presence of the Cas9 gene by PCR. 

##### Cell Sorting to Analyze Transfection Efficiency and Cell Survival 

We nucleofected 800 ng of pX458, 800 ng of GFP construct, and 500 ng of pMAX in 2 × 10^5^ NB4-Cas9 and HL60-Cas9 cells. After 24 h of nucleofection, the cells were stained with 7-Aminoactinomycin D (7-AAD; BD Biosciences). The rates of the GFP-positive transfected cells were measured by flow cytometry using the BD fluorescence-activated cell sorter (FACS) Canto II system. Apoptotic cells (7-AAD^+^) were excluded from the analysis.

## 3. Results

### 3.1. Target-Specific sgRNA Expression Constructs Created by Fusion PCR

To develop an alternative strategy for genetically editing hard-to-transfect leukemic cell lines, we first designed a DNA vector to enable the delivery of small sgRNAs into these cells. Specifically, we engineered the pEGR1 cloning vector, which retained the sequence of the sgRNA expression cassette (including the pU6 promoter, sgRNA scaffold sequence, and the terminal SV40 Poly(A) signal) that was cloned from the parental PX458 plasmid (Figure 1A) into the pEGFP-N1 vector.

From the pEGR1 plasmid, it is possible to generate two types of constructs using fusion PCR. Fusion PCR can be used to produce fused DNA fragments, obviating the need for the restriction of enzyme digestion and DNA ligation. In this scenario, we employed different oligonucleotide primers, with overlapping sequences to produce DNA constructs. Our pEGR1 cloning vector then served as a scaffold for two types of constructs produced through PCR: the first one carrying the sgRNA for *MYBL2*, *IDH2*_1, or *IDH2*_2 (465 bp) and the second containing the CRISPR/Cas9 guide and EGFP reporter (2033 bp). A schematic illustration of the constructs of pU6-sgRNA and EGFP-U6-sgRNA generated through fusion PCR is presented in Figure 1B.

### 3.2. Gene Editing in HEK293 Cells

Prior to transfecting the leukemic cells, we assessed the functionality of the fusion-PCR-derived sgRNA expression constructs in the HEK293 cells, using different plasmid/construct ratios (Table 2). We tested different combinations of hCas9 and sg*MYBL2* constructs to optimize the target-specific cleavage: 500 ng hCas9 and 23.3 ng sg*MYBL2*; 250 ng hCas and 12 ng sg*MYBL2*; 250 ng hCas9 and 35 ng sg*MYBL2*; 250 ng hCas9 and 60 ng sg*MYBL2*; and 150 ng hCas9 and 7 ng sg*MYBL2*. To test the effect of the 5’ termini phosphorylation, the phosphorylated sg*MYBL2* construct was transfected together with the hCas9 vector. Finally, we assayed the transfection of 35 ng of the sg*MYBL2* PCR construct along with 250 ng of the PCR-amplified Cas9 cassette. For *IDH2* editing, 35 ng of both guides (17.5 ng of each guide) was co-transfected with 250 ng of the hCas9 vector. For the R172K mutation editing experiments, 10 µM of ssODN was used. The reproducibility of the experiments was guaranteed because we performed each experiment three or more times (see the tables and figures).

We found that augmenting the quantity of transfected DNA resulted in a dose-dependent increase in the NHEJ DNA repair pathway. As the maximum NHEJ efficiency (10.3 ± 2.2%) was achieved by combining 250 ng of hCas9 with 35 ng of sg*MYBL2* and the ICE analysis indicated no disparities between the different concentrations (Appendix A), we opted for 35 ng and 250 ng for the subsequent experiments. The PX458 plasmid with the sg*MYBL2* guide produces similar results, confirming the effectiveness of the matched MYBL2 guide (7.8 ± 2.5%) using the T7E-I assay (Table 2) and the 11.5 ± 1.3% through ICE analysis (Appendix A).

We then transfected the sg*MYBL2* construct with 5′ phosphorylated ends (which has been reported to enhance expression [30]) into the HEK293 cells. However, no significant differences in NHEJ efficiency were observed between the non-phosphorylated constructs based on the T7E-I assays and the ICE analysis (Table 2 and Appendix A). 

We reasoned that utilizing a smaller construct (encoding only the Cas9 protein) rather than the standard CRISPR plasmid (which includes additional plasmid replication components) would enhance genome integration. As a result, we isolated and amplified the Cas9 expression cassette (including the CMV promoter, Cas9 gene, and HSV TK poly (A) signal) from the hCas9 plasmid by using PCR. Although the amplicon for the isolated cassette was roughly half the size of the entire plasmid (~5000 vs. 9553 bp) and demonstrated functionality in the HEK293 cells, the efficiency of the NHEJ DNA repair was substantially reduced (4.1 ± 1.4%) in comparison to when the hCas9 or PX458 plasmid backbones were used (Table 2). This observation was also confirmed by the ICE analysis results (Appendix A).

Considering the previous optimization efforts, we proceeded to investigate the construct functionality by targeting the *IDH2* gene, utilizing sg*IDH2*_1 and/or sg*IDH2*_2 (Figure 2A). We co-transfected HEK293 cells with 250 ng of the hCas9 plasmid combined with 35 ng of expression constructs. The co-transfection of both sgRNA scaffolds resulted in the highest NHEJ efficiency (21.1± 3.6%) compared to when each sgRNA was used alone (10.8 ± 3.1% for sg*IDH2*_1 vs. 8.3 ± 2.2% for sg*IDH2*_2) (Figure 2B and Table 3). However, the co-transfection of both scaffolds with the PCR-amplified Cas9 cassette led to a reduction in NHEJ by 3.36 ± 0.55% based on the T7E-I assay (Table 3). Similar results were obtained with the ICE analysis (Appendix A). 

Finally, to introduce the IDH2^R172^ mutation into the HEK293 cells, we transfected 10 µM of ssODN (carrying the R172 point mutation, six silent modifications to avoid further DNA cleavage events, and a restriction site for HhaI (Figure 3)) along with the hCas9/*IDH2* expression construct. The HDR efficiency was found to be 1.01 ± 0.19% (Table 3).

### 3.3. Compatibility and Gene Editing in Leukemia Cell Lines

To evaluate the versatility of our sgRNAs using commercially available CRISPR plasmids and their compatibility with different leukemic cell lines, we nucleofected EGFP-sg*IDH2*_1 in combination with EGFP-sg*IDH2*_2, EGFP-sg*MYBL2*, PX458-*MYBL2*-1, or the pMAX plasmid (used as a positive control) into our NB4-Cas9 and HL60-Cas9 cell lines. Then, the live EGFP-positive cells were quantified by flow cytometry with 7-AAD staining. The cell viability following basic nucleofection (without any plasmid/construct) was 68.7 ± 11.3% for the NB4-Cas9 cells and 54.5 ± 6.7% for the HL60-Cas9 cells. The cells nucleofected with plasmids and/or constructs showed similar survival rates (Figure 4). The effects of nucleofector-induced lethality are depicted in Figure 4A,B. As expected, transfection with only the PX458 plasmid (negative control) yielded less than 1% live EGFP-positive cells. Our *IDH2* fluorescent reporter construct revealed 12.5 ± 4.1% live EGFP-positive NB4-Cas9 cells and 7.1 ± 3.8% live EGFP-positive HL60-Cas9 cells, while our *MYBL2* fluorescent reporter construct revealed 10.2 ± 1.9% and 7.2 ± 3.3%, respectively. The highest activity was observed with the pMAX plasmid (positive control), resulting in 22.4 ± 6.2% NB4-Cas9 and 17.9 ± 6.8% HL60-Cas9 EGFP-positive cells (Figure 4).

Then, we focused on *IDH2*-specific gene targeting and editing in NB4-Cas9 cells. First, we transfected each of the sg*IDH2* scaffolds at two different equimolar concentrations (800 and 1500 ng). Although the differences were not statistically significant, we observed a slightly higher efficiency of NHEJ DNA repair (14.52 ± 7.7%) when 1500 ng was used (Figure 5A). The ICE analysis yielded 4.6 ± 2.9% NHEJ efficiency (Appendix A). Furthermore, we optimized the ssODN concentration to 100 µM, which resulted in a 2.2 ± 0.4% HDR efficiency (Figure 5B and Table 4). These conditions were replicated using 750 ng of the sg*MYBL2* expression constructs in the NB4-Cas9 cells, yielding a 2 ± 1.1% NHEJ efficiency. Similarly, the mean gene editing efficiency through the NHEJ DNA repair pathway was determined as 3.6 ± 0.88% in *IDH2* and 6.3 ± 3.3% in *MYBL2* in the HL60-Cas9 cells (Table 5). Nevertheless, the introduction of the IDH2^R172^ mutation was not detected. 

As our method for producing sgRNAs was shown to be functional in the NB4-Cas9 cells, we proceeded to compare the efficiency of these PCR-generated guides with the RNP complexes. For this objective, we focused on the *IDH2* gene. The transfection of the RNP complexes into the NB4 cells yielded 29.8 ± 3.8% NHEJ (Figure 5C). On the other hand, the ICE algorithm analysis resulted in 74.3 ± 3.6% NHEJ efficiency (Appendix A). Finally, with ssODN, an average HDR efficiency of 2 ± 0.3% was achieved (Table 4) (Figure 5D), which is comparable to the efficiency of our method for producing CRISPR components.

### 3.4. Deep Sequencing of CRISPR-Treated Cells Validates Efficacy and Specificity of Our Fusion PCR-Generated Constructs

To confirm the efficiency of CRISPR-mediated gene editing and potential off-target modifications among the population of NB4-Cas9 and NB4 cells edited for IDH2^R172^ mutation, we conducted amplicon deep sequencing. The sequencing data were analyzed using the CRISPResso 2 software. The most edited reads obtained are detailed in Figure 6. After classifying the reads by the type of editing, we found that for the NB4-Cas9 edited cells, 10.95% of the reads showed NHEJ repair and 22.37% was shown for the NB4 edited cells. The predominant pattern in the edited NB4-Cas9 cells showed deletions in the range of 20–29 bp, followed by additional deletions of 30–39 bp in the edited NB4 cells. The prevalence of reads with precise DNA deletion between each PAM sequence is a characteristic finding of the combined use of two sgRNAs [31]. 

In the edited NB4-Cas9 cells, a total of 2.44% knock-ins was observed, including 0.42% of the reads with all the ssODN changes and excluding the single-base *IDH2*_2 PAM modification. These events successfully eliminated the HhaI restriction site. Alternatively, 1.64% of the reads were edited in the standard NB4 cells. CRISPResso2 classified reads with ssODN changes, indels, or deletions as partial (or imperfect) HDRs accounted for less than 0.5% of the reads. Finally, 0.19 and 5.59% of the reads were categorized as ambiguous for the NB4-Cas9 and NB4 cells, respectively. These reads showed deletions ranging between 22 and 99 bp, causing the software’s quantification to be altered and failing to meet the criteria for the NHEJ repaired reads.

Finally, if we consider all the altered reads as Cas9 activity, the NGS findings align with the results of the T7-EI assays. In fact, NGS identified a 14.52% frequency of CRISPR-induced mutations caused by NHEJ and 13.82% caused by NHEJ in the NB4-Cas9 cells. Similarly, in the standard NB4 cells, NGS and the T7-EI assay detected 29.9% and 29.8% mutations caused by NHEJ DNA repair, respectively. A negligible percentage of altered reads was detected in the sixteen potential off-targets of this massive sequencing analysis, further confirming the safety of this technology.

## 4. Discussion

The CRISPR/Cas9 technology has brought about a revolution in hematology by paving the way for the development of novel in vitro and in vivo research models. This has, in turn, led to the exploration of alternative therapies [10]. Traditionally, lentiviruses have served as a vehicle for delivering the CRISPR machinery, especially in in vitro experiments with hematopoietic cells. For instance, among leukemia models, the reported frequencies of CRISPR-induced mutations due to NHEJ DNA repair ranged from 10 [14] to 90% [11], while HDR efficiency was <10% when a DNA template was used for recombination [14]. The advantages of this technology include the utilization of CRISPR/Cas9 lentiviral libraries for the identification of novel drug targets [32], previously unknown tumor suppressor genes [33], and the mechanisms involved in cytarabine resistance [34] in AML models. However, the lentiviral integration and constitutive expression of CRISPR/Cas9 elements may increase the chances of undesirable effects, such as off-target events resulting from excess sgRNA or tolerable mismatches between target regions and the PAM sequence [16].

To address the drawbacks of lentiviral transfection, we developed a “hit-and-run” strategy to efficiently edit specific genes in leukemic cells, requiring only a single lentiviral insertion event. We first developed the pEGR1 reporter expression cassette and a streamlined fusion PCR protocol to merge specific sgRNAs, targeting the *MYBL2* and *IDH2* genes. We assessed hCas9-mediated editing efficiencies in HEK293 cells through co-transfection with our fusion-PCR-generated sgRNA. This approach produced a higher tendency for NHEJ editing within the target sequence compared to the standard CRISPR plasmids (i.e., PX458) used for *MYBL2* gene targeting in the HEK293 cells, thus simplifying the process. Moreover, our system substantially improves transfection and, therefore, gene targeting. In agreement with previous reports [35], the utilization of two sgRNAs increases CRISPR efficiency.

Although the frequencies of CRISPR-induced mutations obtained in the HEK293 cells were moderate, our results are in line with previously published data [36,37,38], even though certain groups reported higher efficiencies in the same cell line [31,39,40]. 

The NB4 and HL60 cells posed challenges for transfection with commercially available CRISPR vectors (i.e., the PX458 or pMAX plasmids). According to the established nucleofection protocols, transfection efficiencies of nearly 80% and 60%, respectively, can be obtained when using the pMAX plasmid (under ideal conditions) in NB4 and HL60 cells. In this study, despite several optimizations, we achieved maximum transfection rates of 22% and 18%, respectively. Nonetheless, these values fall within the range of editing efficiencies in other leukemic cells. For instance, the K562 cell line, which models chronic myelogenous leukemia, is often used for optimizing gene editing protocols for stem cells due to its high transfection rate [11,39]. In our study, however, we used lentiviral transduction to generate stable cell lines that constitutively expressed the Cas9 transgene, thereby avoiding the subsequent use of viral vectors. We created two leukemic cell models (i.e., NB4-Cas9 and HL60-Cas9) that can serve as a base for CRISPR transfections in future investigations. 

The use of RNPs is recommended when potential off-targets have been predicted since they have a reduced window of activity. In agreement with previous reports [41], our experiments showed that RNPs increased CRISPR efficiency, albeit without a corresponding increase in HDR rates. This finding supports the previous data, indicating that although HDR ensures reliable replication by using the sister chromatid as a template for DNA repair [42], NHEJ restores genome integrity more quickly (at the cost of producing more errors) and remains the preferred DNA repair pathway for cells [43]. 

Using flow cytometry, we detected higher transfection ratios with our EGFP-expressing constructs compared to the PX458 (~10 kb) plasmid in the NB4-Cas9 and HL60-Cas9 cells. These findings align with the work of Wu et al., who showed that PCR-amplified EGFP produces higher nucleofection efficiency in NB4 cells than its parental plasmid, with minor effects on cell viability [44]. Applying our methodology, specific targeting of the *IDH2* gene in the NB4-Cas9 cells yielded 10.5% and 2.2% frequencies of mismatched mutations caused by the NHEJ and HDR DNA repair pathways, respectively. Meanwhile, the fusion-PCR-generated constructs targeting *MYBL2* resulted in a 2% frequency of mismatched mutations caused by NHEJ. To assess the versatility of our strategy, we replicated the same methodology in the HL60-Cas9 cell lines, achieving 3.6% mutagenesis in *IDH2* and 6.3% in *MYBL2*, both caused by the NHEJ DNA repair. The efficiency of different transfection vectors depends on multiple variables, all of which affect the resulting reproducibility.

Though targeted NGS, we confirmed the NHEJ and RFLP results obtained by using T7 and RFLP in IDH2^R172^-edited NB4-Cas9 cells. Moreover, we examined 16 potential off-targets in edited NB4-Ca9 with our constructs and NB4 cells edited with RNPs, with no off-target effects detected. This corroborates our idea that the Cas9 gene inserted into the NB4-Cas9 genome is secure.

## 5. Conclusions

Our methodology has expanded the CRISPR/Cas9 technology toolkit by developing a robust and cost-effective strategy for generating CRISPR constructs through fusion PCR. While we demonstrated this proof of concept using HEK293, NB4, and HL60 cell models, the versatility of this technology enables its application in various cell types.

## Figures and Tables

**Figure 1 cancers-15-04263-f001:**
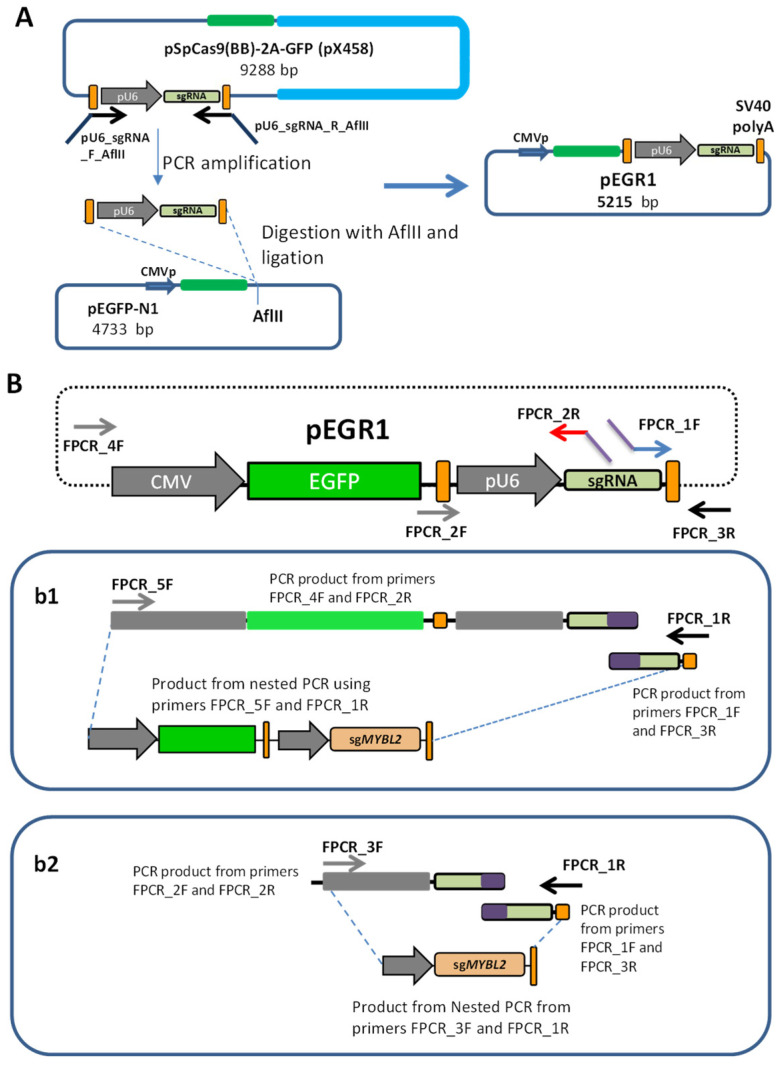
Fusion-PCR-generated sgRNA expression constructs. (**A**) Development of the pEGR1 vector as a template to generate sgRNA constructs by PCR. The entire sgRNA cassette was amplified from the pX458 plasmid, using pU6_sgRNA_F_AflII and pU6_sgRNA_R_AflII primers and was then inserted into the pEGFP-N1 vector to generate the pEGR1 plasmid. (**B**) Schematic illustration of the pEGFP-pU6-sgRNA scaffold plasmid, pEGR1. Specific primers were used for the fusion PCR to generate a construct containing either the EGFP cassette together with the sgRNA (**b1**) or just the sgRNA (**b2**). To create these constructs, we used the PCR_1F and PCR_3R primers to amplify the terminator sequence (part of the coding RNA guide plus the SV40 polyA terminator) with a specific sequence of the target (purple); this was then used for both constructs, (**b1**,**b2**). The FPCR_1F primer contains a specific sequence to the target locus of 20 bp in the 5’ end (purple) that provides specificity to the RNA guide. The common reverse FPCR_2R primer will be used for both constructs (**b1**,**b2**) and needs to contain the same specific sequence introduced in FPCR_1F (purple tail), but reversed and complementary so that they perfectly overlap during nested fusion PCR (see below). These specific sequences (purple bit) need to be introduced by the researcher to target their locus of interest (**b1**). To produce the whole CRISPR construct containing the EGFP cassette plus the RNA guide, we used the common end of the construct produced by the FPCR_1F and PCR_3R primers, together with a PCR product produced from pEGR1 using primers FPCR_4F and FPCR_2R. The primer FPCR_2R contains a 20-nucleotide specific sequence, reversed and complementary to the tail of the FPCR_1F primer used to produce the common end PCR product. Then, we mixed both PCR products and used the internal FPCR_5F and FPCR_1R primers to produce the nested fusion PCR final product (**b2**). To produce the small construct containing just the RNA guide, we used a similar strategy but created a 5′ PCR product that only contained the pUC6 promoter and the rest of the sequence required to express the RNA guide, using the primers FPCR_2F and FPCR_2R. Then, we performed a nested PCR using FPCR_3F and FPCR_1R.

**Figure 2 cancers-15-04263-f002:**
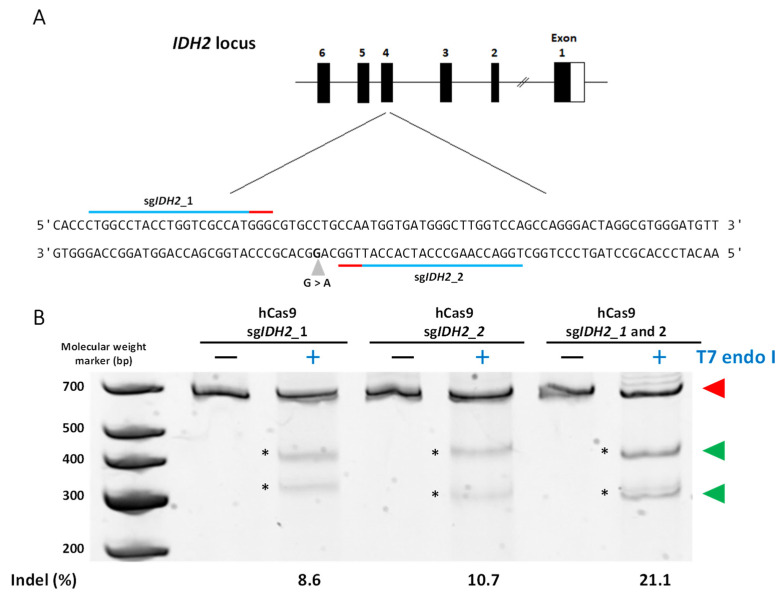
Targeting *IDH2* gene in HEK293 cells. (**A**) Structure of the *IDH2* gene, depicting the position of the R172K mutation in exon 4, caused by a G to A change (grey arrowhead). The sequence over which the sgRNAs were designed is indicated by the blue lines, while the red lines indicate the protospacer adjacent motif (PAM) sequences. (**B**) The NHEJ efficiencies were obtained after analysis of the intensity of the bands, which are products of the T7 endonuclease I digestion. The assay shows lanes for each candidate sgRNA. The percentages shown underneath the gels represent the relative intensities of the bands (%). The red triangles point to the undigested products, while the green triangles point to T7E-I products. At the left of the gel, the bands from the GeneRuler Low Range DNA Ladder (ThermoScientific) appear. The original nucleic acid electrophoresis images can be found in Appendix A.

**Figure 3 cancers-15-04263-f003:**
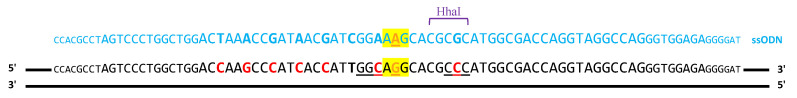
Design of the ssODN donor template with *IDH2*^R172^ mutation. A portion of the sequence of the *IDH2* is presented. The position of the R172 mutation is indicated by the orange nucleotides (A to G substitution, which results in R to K change in the enzyme. The codon is highlighted in yellow), whereas the seven silent mutations introduced to avoid Cas9 reiterated cleavage are indicated by the red nucleotides. The changes introduced into the PAM sequence (underlined nucleotides) of the sgRNA_2 correspond with the HhaI target sequence on the ssODN, which is in turn used to detect the mutations resulting from HDR, using the RFLP analysis.

**Figure 4 cancers-15-04263-f004:**
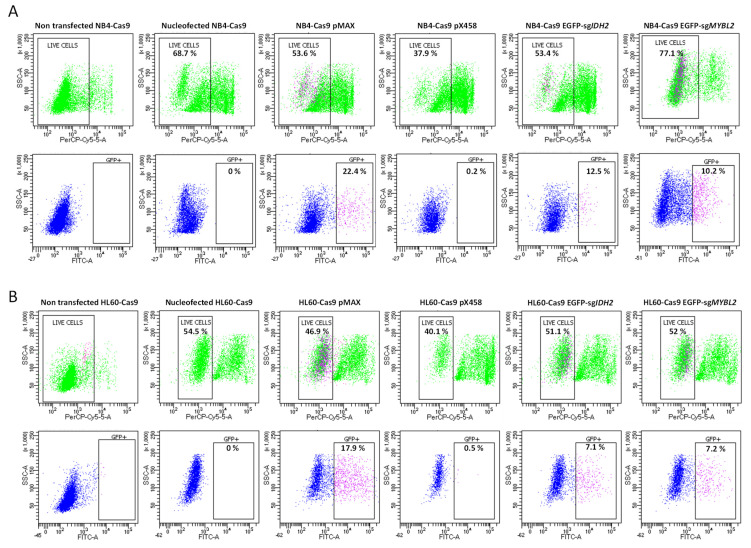
Analysis of the efficiency of cell internalization and cell survival of PCR-generated sgRNAs in NB4 and HL60 cells. Representative results of flow cytometry with numbers indicating the percentages of live cells and live GFP-positive cells. NB4-Cas9 (**A**) and HL60-Cas9 cells (**B**) transfected with EGFP-sg*IDH2* and EGFP-sg*MYBL2* constructs show a higher proportion of EGFP-positive cells (indicated in pink) than cells transfected with the pX458 plasmid. Non-transfected cells and cells nucleofected without any plasmids were used as controls to study lethality. Cells nucleofected with pMAX plasmid were used as positive transfection control.

**Figure 5 cancers-15-04263-f005:**
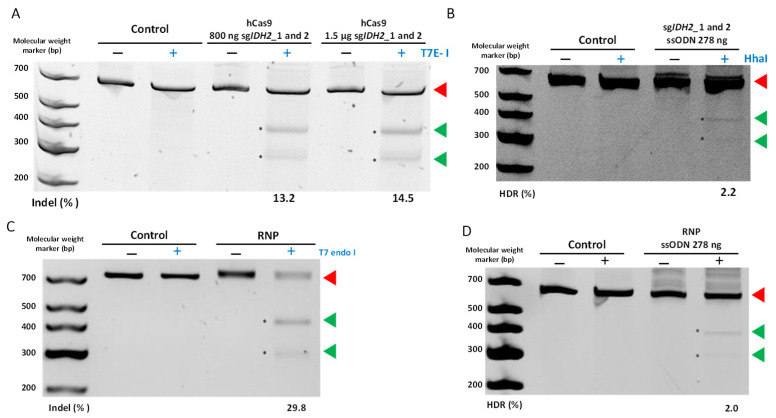
Detection of CRISPR-mediated editing in *IDH2* in NB4-Cas9 cells. The PCR products of the T7 endonuclease I assay and RFLP analysis were separated by polyacrylamide gel electrophoresis, respectively, to detect the indels created as a result of the non-homologous end joining (NHEJ) DNA repair and HDR efficiency, following transfection of (**A**) hCas9 with either 800 or 1500 ng of sgRNA; (**B**) 100 µM of ssODN as a template for HDR; (**C**) ribonucleoprotein (RNP) complexes; (**D**) RNP complexes in addition to 100 µM of ssODN as a template for HDR. The cleaved products (indicated by the asterisks) were used to quantify indels created by NHEJ or HDR efficiency. The numbers beneath the gels represent the relative band intensities (%). Following quantification, the brightness and contrast of the gel image in (**B**) were modified for better visualization. Red triangles point to undigested products, while green triangles point to digested products. The original nucleic acid electrophoresis images can be found in Appendix A.

**Figure 6 cancers-15-04263-f006:**
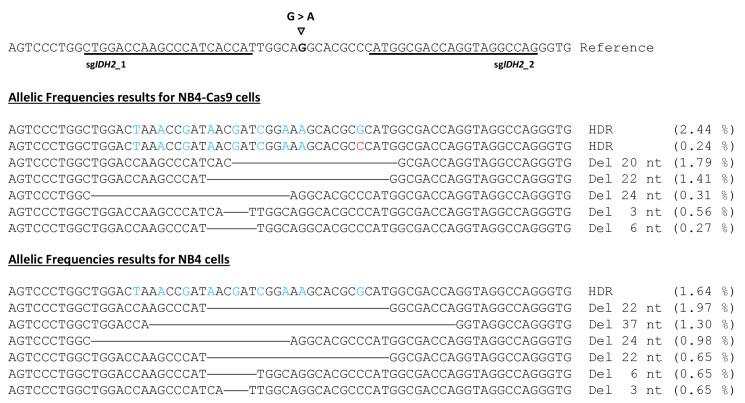
Summary of the main CRISPResso2 results identifying alleles generated from the editing of the *IDH2* gene. The reference sequence with the selected sgRNA guides (underlined) and the desired single-base substitution. Deleted regions are indicated by horizontal lines. Blue nucleotides result from HDR based on the ssODN template. The red nucleotide represents an error caused by imperfect HDR DNA repair.

**Table 1 cancers-15-04263-t001:** Off-targets selected.

	Chromosomal Position	Gene Name	Sequence	Region
sg*IDH2*_1				
1_off1	chr3: 76082901	*ROBO2*	cGGACCAAGgCgATCACCATGGG	Intron
1_off2	chr4: 130711158 (-)		TGtACCAAGCtCATCAaCATTGG	Intergenic
1_off3	chr1: 2405874	*PEX10*	TGGgCCAtGCCCATCcCCATCGG	Intron
1_off4	chr1: 54465152		TGGACCAAGCCCcTCACCtTGGG	Intergenic
1_off5	chr15: 70078666	*TLE3*	TGGcCCAAGCCCtTCACCAaCGG	Intron
1_off6	chr17: 83115842 (-)		TGGACCAAGtCCAcCtCCATGGG	Intergenic
1_off7	chr10: 127909863 (-)	*PTPRE*	TGGACCAtGCCCATCcaCATCGG	Intron
1_off8	chr6: 31629230	*PRRC2A*	gGGACCAAtCCCATCACCcTTGG	Exon
1_off9 ^1^	chr14: 99534805	*CCNK/CCDC85C*	TGGcCaAAGCCCtTCACCATAGG	Exon/Intron
1_off10	chr9: 38523669	*RP11-103F21.4*	gGGACCAgGCCCtTCACCATTGG	Pseudogene
1_off11	chr9: 97112588		gGGACCAgGCCCtTCACCATTGG	Intergenic
1_off12	chr18: 58014733 (-)		TGaACCAAGCCCATaACCcTTGG	Intergenic
sg*IDH2*_2				
2_off1	chr5: 173428070	*CTB-32H22*	CTGaCCTgCCTGGTCcCCATTGG	Intron
2_off2	chr1: 205830643 (-)	*PM20D1*	tTGGCCTcCCTGGTCGtCATGGG	Intron
2_off3	chr2: 241745044	*D2HGD*	CTGGCCTtCCTGGTgGtCATGGG	Intron
2_off4	chr17: 2879971(-)	*RAP1GAP2*	CaGGCCTACCTGGTCcCCATTGG	Intron

^1^ Off-target 1.9 sequence corresponds with the last exon of *CCNK* gene and the first intron of *CCDC85C* gene. Mismatches between sgRNA and off-target sequence are indicated in lowercase nucleotides.

**Table 2 cancers-15-04263-t002:** Results of NHEJ obtained in the different optimizations probed with sg*MYBL2* in HEK293 cells.

Conditions	150 ng hCas9 7 ng sg*MYBL2*	250 ng hCas9 12 ng sg*MYBL2*	250 ng hCas9 35 ng sg*MYBL2*	250 ng hCas9 60 ng sg*MYBL2*	500 ng hCas9 23.3 ng sg*MYBL2*	250 ng hCas9 35 ng sg*MYBL2*-P	250 ng PCR Cas9 35 ng sg*MYBL2*	PX458-*MYBL2*
Exp.1	4.5	3.4	5.2	6.0	5.2	14.0	3.0	4.3
Exp. 2	3.0	5.8	8.0	6.0	3.0	8.0	7.0	11.0
Exp. 3	3.7	4.0	13.0	14.7	14	6.0	2.4	3.0
Exp. 4	2.5	4.0	14.8	5.3	8.4	13.9	-	13.0
Average NHEJ (% ± SEM)	3.4 ± 0.4	4.3 ± 0.5	10.3 ± 2.2	8.0 ± 2.2	7.6 ± 2.3	10.5 ± 2.0	4.1 ± 1.4	7.8 ± 2.5

**Table 3 cancers-15-04263-t003:** NHEJ and HDR results obtained for *IDH2* addition optimization in HEK293 cells.

Conditions ^1^	sg*IDH2*_1	sg*IDH2*_2	sg*IDH2*_1 + sg*IDH2*_2	PCR Cas9 sg*IDH2*_1 + sg*IDH2*_2	sg*IDH2*_1 + sg*IDH2*_2 ssODN ^2^
Exp.1	19.0	7.1	24.2	4.4	0.5
Exp. 2	11.0	15.0	28.8	2.8	0.8
Exp. 3	9.0	5.4	11.8	2.8	1.1
Exp. 4	4.0	5.5	19.7	-	-
Average NHEJ (% ± SEM)	10.8 ± 3.10	8.3 ± 2.20	21.1 ± 3.60	3.36 ± 0.55	
Average HDR (% ± SEM)					1.01 ± 0.19

^1^ For each experiment, 250 ng of hCas9 vector and 35 ng of guide construct were used. We used the same amount of DNA for the experiment using PCR-amplified Cas9 (i.e., 250 ng). ^2^ In every experiment, 27.8 ng of ssODN was used for homologous recombination.

**Table 4 cancers-15-04263-t004:** NHEJ and HDR efficiencies using PCR constructs in NB4-Cas9 cells and RNPs in NB4 cells.

Conditions ^1^	sg*IDH2*_1 + sg*IDH2*_2 ^2^	RNPs ^3^	sg*IDH2*_1 + sg*IDH2*_2 ^2^ + 278 ng ssODN	RNPs ^3^ + 278 ng ssODN	sg*MYBL2* ^2^
Exp. 1	21.5	28.7	2.8	2.5	4.0
Exp. 2	7.0	34.16	2.3	1.6	2.0
Exp. 3	15.0	26.7	1.5	2.0	0
Average NHEJ (% ± SEM)	14.5 ± 7.7	29.8 ± 3.8			2 ± 1.1
HDR (% ± SEM)			2.2 ± 0.4	2.0 ± 0.3	

^1^ For editing both genes, 750 ng of each sgRNA construct was used. ^2^ PCR-generated guides. ^3^ RNP means ribonucleoproteins containing both *IDH2* guides, synthesized by IDT-DNA. When using RNPs we used NB4 naïve cells, and the rest of the experiments were performed on NB4-Cas9 cells.

**Table 5 cancers-15-04263-t005:** NHEJ efficiencies using PCR constructs in HL60-Cas9 cells.

Conditions ^1^	sg*IDH2*_1 + sg*IDH2*_2 ^2^	sg*MYBL2* ^2^
Exp. 1	2	8
Exp. 2	5	11
Exp. 3	4	0
Average NHEJ (% ± SEM)	3.7 ± 0.9	6.3 ± 3.3

^1^ For editing both genes, 750 ng of each construct was used. ^2^ PCR-generated guides.

## Data Availability

The fastaq data obtained files are public on the European Nucleotide Archive, under the project number PRJEB54077.

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
