# Peer review of "PCR-Based Strategy for Introducing CRISPR/Cas9 Machinery into Hematopoietic Cell Lines"

_cancers, 2023, doi:10.3390/cancers15174263_

Round 1
Reviewer 1 Report
The manuscript of González-Romero and coauthors describes an application of genome editing for two hematopoietic cell lines exemplified for two target genes. It is commonly known, that hematopoietic cells are difficult to modify. Therefore, stable viral transduction has been establish as standard method for the manipulation of this cells line. Since the packaging capacity of viral particles is limited, lentiviral systems are preferable used over adenoviral systems. In addition, separation of transduction of components for multicomponent systems has been established. The authors split the transduction of the large bacterial Cas9 gene from the transfection of sgRNA’s, a common method for the application of gene editing systems. Differences of efficiencies of different systems used were marginal. Due to the low reproducibility no general conclusions can be drawn. Thus, the manuscript is of limited interest. Still, the combinatory approach of different techniques for cell editing as well as cell characterization might be of interest for people working with difficult to modify cell systems.
Before the manuscript can be recommended for publication, a major revision is necessary to improve its quality. It is superficially written and contains numerous typos. Major concerns are the content and structure of the material and methods section, the layout and quality of all figures and tables as well the discussion. Supplementary material need to be links in the main text. In addition, typos need to be excluded from the manuscript.
The material and method section needs to structured in materials used and methods. Titles of particular methods need to descriptive. I would recommend to separate cloning work from cell experiments. Methods need the comprehensibly described. Typos should be eradicated.
Lane 168 ff (“Optimization of …”) Here a method should be described, not a particular experiment.
Lane 183: Which “PCRs” were used, specify!
Lane 188: Briefly introduce the T7-EI assay here.
Lane 198: What is a “ICE“?
Lane 207; 214 etc.: Numbers are not appropriately written.
Lane 210: What is Cl2Ca? Correct!
Lane 227: The formation of the RNP does not belong to Nucleofection. Briefly introduce the system to form RNPs.
Lane 245: A CANTOTM does not exist. Correct.
Throughout the results section reproducibility of the experimental setup need to be specified. All figures need to be improved significantly. Tables are difficult to read and should be improved.
Figure 1: Part B is not understandable. The smaller DNA fragments cannot be clearly linked to primer pairs described of the template vector. Specify. Add the name of the vector.
Table 2 is corrupt and impossible to read. How were the NHEJ efficiencies quantified? The efficiencies of the particular experiments were quite different. Thus, they do not allow to draw conclusion about the efficiency of NHEJ in the particular settings. Differences between particular experiments need to be discussed.
Figure 2: What are the sequences of sgIDH2_3 and 8? Add the size of the molecular weight marker fragments. Which sgRNA sequences were used for the hematopoietic cell lines?
Figure 3: Specify the codon for position R172.
Table 3: Specify NHEJ and HDR samples. Specify amount of nucleic acids used. Why were ones an amount (ng) and ones a concentration (µM) used. This is not comparable. The sentence under the table is not understandable. The particular experiments in the tables 2 and 3 reveal that efficiencies are not reproducible. In general, columns with error bars would reveal better to compare the settings.
Figure 4: Improve the title of the figure legend. The quality is very low. Improve. An information about the reproducibility of the data should be provided. From the FACS dot blots no conclusions about the cell survival can be drawn. What do the yellow events in the figures indicate? The percentage of GFP-positive cells should be added in the figures. Why different gating were used in figure 4A. It should be similar. No information about the transfected nucleic acids is provided. In general, if cell survival an issue for the efficiency of transfection, the authors should provide this information.
Figure 5: Again, amounts of transfects nucleic acid were compared to a concentration. The size of the molecular weight marker fragment should be added. A Figure S2 A-D is not provided.
Table 4: What is the difference between lane 1 and 2 to lane 3 and 4? What was the amount of sgMYBL2 used?
Table 5: The mean calculation of the sgIDH2 is wrong.
Lane 447 ff; Figure 6: Were data reproducible? How many different sequences were obtained in the NGS? Are the presented sequences the most efficient alterations? Surprisingly the efficiencies of allelic frequencies of NB4-Cas9 and NB4 cell were quite similar.
The sequences for sgIDH2_1 and sgIDH2_2 presented in figure 6 are opposite compared to the sequences in figure 2A and different from the sequences in Suppl. Table S4. What is correct?
For NB4-Cas9 cells: Why were two different allelic frequencies for HDR in NB4-Cas9 cell presented?
For NB4 cells: Efficiencies of transfection of NB4 cells are not provided in the manuscript. These data need to be added. It remains to be clarified how sgRNA can act in absence of Cas9?
The discussion if rather long and frequently speculative (i. e. lane 501 -504, lanes 506 - 517). Conclusions cannot be drawn based on the results provided. I would recommend to significantly shorten the discussion.
Lane 498: For proliferating cells the chromatin structures are not too important since the DNA is accessible during cell division.
Lane 519 ff: The manuscript claims that NB4 and HL60 cell are difficult to genetically modify, but the authors state that efficiencies up the 80 % are possible to obtain. The authors obtained about 20 % of efficiency. This is by far still sufficient to obtain cell clones, which are genetically edited.
Lane 571 ff: The use of supplementary material need to be linked to the main manuscript.
Suppl. Table S4: Why was the IDH2-R172 sequence provided in small letters?
Suppl. Table S8: What are indexes?
Typos need to be ommitted
Author Response
Responses to both referees are in the author-coverletter-30901826.v1.docx

Reviewer 2 Report
González-Romero et al. demonstrated in this paper that PCR-generated single guide RNA is useful for gene-editing in two hematopoietic cell lines, NB4 and HL60. Except this, the authors did not address any biological questions in the paper. Therefore, I believe that this paper is more suitable for some journals specialized in methodologies.
Author Response
#Referee 2
We deeply thank referee 2 for being so supportive.
Submission Date 10 July 2023
Date of this review 18 Jul 2023 09:59:34
Round 2
Reviewer 1 Report
The revised manuscript improved significantly. The concerns were addressed appropriately.
Small typos can be processed in the printing process. Size of text parts in the figures has to be adjusted to the printing requirements.
Reviewer 2 Report
If this methodological paper is informative for anybody, it would be O.K. to be published in this journal.